# Impact of COVID-19 on Patients’ Attitudes and Perceptions of Dental Health Services: A Questionnaire Based Study in an Australian University Dental Clinic

**DOI:** 10.3390/healthcare10091747

**Published:** 2022-09-12

**Authors:** Hashim Azam, Niesha Agilan, Pulina Pitigala, Anjaneya Gupta, Julian Fung, Catherine M. Miller, Oyelola Adegboye, Dileep Sharma

**Affiliations:** 1College of Medicine and Dentistry, James Cook University, Cairns, QLD 4878, Australia; 2Australian Institute of Tropical Health & Medicine, James Cook University, Smithfield, QLD 4870, Australia; 3College of Public Health, Medical & Vet Sciences, James Cook University, Townsville, QLD 4811, Australia

**Keywords:** COVID-19, dental health services, public health, questionnaire, attitudes, university dental clinic

## Abstract

COVID-19, the global pandemic, has significantly interrupted the provision of oral health care to many individuals. This study aims to evaluate patients’ attitudes to and perceptions of dental visits in the COVID-19 pandemic and assess if socio-economic status influences their perception of risk associated with dental visits. Patients attending the dental clinic were invited to participate in this study by completing a questionnaire administered in August 2021. Composite indicators for access, attitude, perception and socio-economic status were created based on subsets of questions. A total of 247 completed questionnaires were obtained. Analysis was performed with the perception, attitude and access indicators against the socio-economic status indicator. This study found that there is a statistically significant difference between socio-economic groups and their attitudes and perceptions around dental health care services in the current COVID-19 pandemic. Individuals from lower socio-economic status groups were less influenced by the pandemic. Participants from higher socio-economic status groups were found to be more cautious around COVID-19 and its risks.

## 1. Introduction

The outbreak of the novel coronavirus SARS-CoV-2 (COVID-19) led to worldwide chaos, with the World Health Organization deeming it a pandemic in early March 2020 [1]. SARS-CoV-2 is usually transmitted via droplets associated with breathing, coughing, sneezing or by contaminated fomites, aerosols or fluid from human secretions in symptomatic or asymptomatic patients [1]. Consequently, the high transmissibility of the coronavirus, coupled with the potential symptoms of severe acute respiratory syndrome, led to the rapid closure of many health care settings, such as dental clinics [2].

In a dental setting, the risk of transmission is significantly increased as aerosol and droplet production from dental instruments, coupled with patient’s secretions, saliva, and blood, can potentially lead to an increased risk of pathogenic microorganisms’ spread [3]. Hence, dental practices were perceived to have a high risk of transmission between dental practitioners and patients. With the heightened risk of coronavirus transmission, patients’ perceptions about attending dental clinics can change significantly [1,2,4]. Furthermore, patient priorities and concerns are likely to differ depending on backgrounds and past experiences.

The health belief model suggests that health behaviours, such as attending dental appointments, are controlled by individual perceptions, modifying factors, and the likelihood of action [4]. Beliefs may also be impacted by the perception of an illness, the importance of one’s health, perceived susceptibility, and perceived severity [1,2,4]. Mainly, an increase in perceived susceptibility to coronavirus in the dental clinic may influence patient attendance, which could be very detrimental to the patient’s long-term dental health [1,2,5]. During the pandemic, the forced shutdown of dental clinics notably delayed access to necessary dental treatment [1,2,4]. This can be detrimental since poor oral health can worsen other systemic diseases such as atherosclerosis, low birth weight and pulmonary disease, as the periodontal biofilm act as a reservoir for pathogenic bacteria [2,6]. Additionally, lack of treatment of pulpitis can result in irreversible pulpitis, causing severe pain, consequently leading to dental abscesses, which can, in some instances, develop into cellulitis, with an increased risk of respiratory obstruction or septicemia [1,2,6]. However, the patient’s attitude and awareness about their oral health are vital to whether the treatment plan will be completed successfully [1,2,6].

Government-funded dental clinics in Australia provide services to low socio-economic groups and old age pensioners [1,2,6]. These individuals can have diverse views on health, which can invariably affect their perception of their dental health [6]. A recent study performed a limited analysis of the role of demographics in perceptions towards dental health in such populations; therefore, this research aims to explore the relationship between socio-economic status and COVID perception [6]. This study also enabled the gathering of data on patient concerns regarding COVID-19 spread. The results of this study could be invaluable in the provision and improvement of dental health care services in the university dental clinic. In the event of further COVID-19 outbreaks, having knowledge of approaches that can improve patient attitudes towards dental care could help enhance patient attendance and oral health compliance.

## 2. Materials and Methods

### 2.1. Study Design, Participants and Data Collection

This cross-sectional study was carried out at James Cook University Dental Clinic, Cairns, Australia, over a three-week period in August–September 2021. Cairns is located in far north Queensland, a region that experienced low case numbers and a minimal impact of the virus while other regions of Australia were under lockdown. Patients who were at least 18 years old and attending the dental clinic during the study period were invited to participate. Patients who had only started receiving treatment at the university clinic after July 2020 and those who could not consent (e.g., minors, patients with a disability affecting their informed decision) were excluded. The study was designated low risk and was approved by the Human Research Ethics Committee of James Cook University (approval number H8487). The risk was mitigated by obtaining verbal and written consent from the patients. Patients had the ability to withdraw their participation at any time.

### 2.2. Instrument

A paper-based questionnaire was designed to obtain data from patients regarding their demographics, socio-economic status, perceptions, access, and attitudes. The design and questionnaire selection were aided by past research conducted by Moffat et al., with modifications made to customise the questionnaire to the demographics of the Cairns region and university dental clinics [6]. The design of the questionnaire was such that questions 1–8, 16 and 19–25 required a multiple-choice answer and questions 9–15, 17–18 and 26–30 had 5-point Likert scale answers (Appendix A). Socio-economic markers were selected based on Kuppuswamy’s socio-economic scale; the data points were employment status, annual income and education level [7].

### 2.3. Data Analysis

Characteristics of the study participants were presented as frequencies, percentages, weighted mean, and standard deviations. Psychometric properties of the survey instrument were assessed using Cronbach’s alpha and Spearman correlation analysis to examine the internal consistency reliability and criterion (concurrent) validity. [8] The Mann–Whitney U-test and non-parametric one-way analysis of variance (ANOVA) were used to compare continuous variables, and the chi-square test was used to compare categorical variables. The Kruskal–Wallis test was used to compare differences between each domain’s perceptions, attitudes, and access and demographic variables. Multiple regression analysis was used to investigate the effect of a patient’s socio-demographic characteristics on each of three domains, perception, access and attitude, separately.

All statistical analysis of the data was carried out using SPSS version 28 (IBM Corp, Armonk, NY, USA) and R version 4.0.1 (The R Foundation, Boston, MA, USA) (The inference was based on a 5% level of significance.)

## 3. Results

A total of 266 participants volunteered to participate in this study, with 247 completing the questionnaire in full. Table 1 describes the socio-demographic characteristics of the patients. Among the 247 patients, 125 (50.6%) were females, and 122 (49%) were males, with more than one-third (36.9%) being elderly, aged 70 years or more. Overall, 105 (42.5%) patients reported attending the dental clinic for more than four years. The highest level of education attained for most of the patients was year 10 (35.6%), and more than half (51.4%) of the patients reported an annual income of less than AUD 30,000. Many of the patients were retired (136, 55.1%), holders of concession cards (184, 74.5%) and non-Aboriginal or Torres Strait Islander (234, 94.7%).

We assessed the level of patients’ perception of risk of COVID-19 exposure, reasons for attending, and feeling of safety while attending a dental clinic during the COVID-19 pandemic (Figure 1 and Figure 2). Most patients, i.e., 170 patients (73.28%), felt the location with the highest risk for COVID-19 exposure was supermarkets, followed by hospitals (31, 13.78%), while the least likely exposure site was dental clinics (Figure 1A). Less than half of the patients (104, 45.6%) reported that the lockdown was the most important hindrance to attending a dental clinic, followed by fear of contracting COVID-19 (42, 18.5%) (Figure 1B). The second most important hindrance was the prioritisation of other commitments, followed by financial problems and lack of transport; fear of contracting the virus was the least important (Figure 1B).

About three-quarters (180) of the patients believed receiving the COVID-19 vaccine should increase their perception of safety while attending a dental practice. Other important actions that positively influenced their perceptions were temperature checks prior to appointments, better social distancing in the waiting room, telephone screening and the availability of hand sanitiser.

A quarter of the participants believed that their dental health had been negatively impacted by COVID-19; of these, 9.7% (*n* = 24) of respondents felt that their dental health was significantly worse (Table 2). Clinic closure due to COVID-19 was perceived to be an appropriate measure by nearly 84.2% of participants. The COVID-19 protocols implemented by the university clinic were deemed to be much the same as other sites by 71.3% of respondents (Table 2). Notably, the introduction of COVID-19 protocols successfully alleviated fears in 43.3% of participants (Table 2).

Travelling to attend the university clinic was deemed to be affected by COVID-19 for 37.6% (*n* = 93) of the patients (Table 3). The difficulty of making a booking was altered for 42.1% (*n* = 101) of participants (Table 3). Nearly 42% (*n* = 101) of individuals reported they had an appointment cancelled due to the COVID-19 closure (Table 3). Just over half of the participants were not notified of the university clinic’s closure due to COVID-19, and just under half were notified of the re-opening of the clinic (Table 3). COVID-19 caused financial hardship for 38.3% (*n* = 88) of participants, and of those individuals, 59.7% (*n* = 40) stated that their financial hardship had made affording dental treatment difficult (Table 3).

In terms of patients’ attitudes towards attendance due to the pandemic, 7.7% (*n* = 19) were slightly less likely to attend, and 11.3% (*n* = 29) were significantly less likely to attend (Table 4). Close to half of the participants stated they were highly likely to attend for a dental emergency with the COVID-19 protocols (Table 4). In the event of a new outbreak of COVID-19, 30.8% (*n* = 76) of patients noted that they were significantly likely to attend, and 12.6% (*n* = 31) indicated that they were significantly less likely to attend (Table 4).

Patients noted that the most important factor in increasing their perception of safety was receiving the vaccine (75.9%), followed by temperature checks prior to appointments, better social distancing in the waiting room, or telephone screening; the least important was hand sanitiser stations (Figure 2).

Internal consistency reliability was checked using Cronbach’s alpha (Table 5). Composite indicators were created for three domains: access, attitudes, and perceptions. The indicators were aggregated using questions in the questionnaire most interrelated to the domain. Questions 16–19 and 21–24 were used to create the access indicator. The attitudes indicator was created using questions 26–30. Answers from questions 9–15 were utilised to create the perceptions indicator. For each indicator, Cronbach’s alpha was checked to ensure internal consistency reliability. The perceptions indicator had α = 0.649, the attitudes indicator α = 0.67 and the access indicator α = 0.607. These were all within an acceptable range, although at the lower limit of acceptability. The socio-economic indicator was significantly negatively correlated with the three domains, access, attitudes, and perceptions, indicating concurrent validity (Table 5).

Statistical tests were performed to investigate differences in each indicator (perceptions, attitudes, and access) against socio-demographic indicators (Appendix B). Our results yielded no significant relationship between most socio-demographic variables (except healthcare/concession card and years at JCU Dental) and attitudes, perceptions, and access to dental services during the COVID-19 pandemic. Figure 3 presents the results from three multiple regressions for each domain indicator of attitudes, perceptions, and access to dental services. Although the test revealed that socio-economic status had a negative effect on all three domains, only the two domain indicators, perceptions and access, were statistically significant after adjusting for other variables in the model. Patients with a higher SES indicator had lower risk perception and access scores. That is why people with a higher SES indicator had a good perception of JCU Dental’s response during COVID-19, in contrast to individuals with a lower SES indicator. Similarly, patients with higher SES indicators were more likely to have had their access to university clinic health services less impacted by COVID-19. The higher SES indicator individuals were more likely to agree with the protocols, believe their dental health was less impacted and have had their fears alleviated by the protocols. Additionally, possession of a health care/concession card was positively and significantly associated with increased attitude and access domains.

## 4. Discussion

This study was conducted to explore the impact of COVID-19 on patients’ perception of dental health services for those who attend a university clinic in a regional Australian town (Cairns). Situated in Cairns, northern Queensland, the local population has largely not been affected by COVID-19, unlike other Australian states such as Victoria and New South Wales. This contrasts with a previous study that analysed patients’ perceptions in the regions with significant COVID-19 cases in the community [6]. This clinic provides dental care to the population in the tropical north Queensland region, including the Tablelands, all of which were not significantly impacted by the COVID-19 pandemic (in 2021). The study aimed to investigate whether the low prevalence of COVID-19 in the region would alter attitudes and perceptions of dental services within the region.

Almost three out of four respondents reported having health care or concession cards, and the SES indicators, which were determined by employment status, annual income and highest level of completed education, showed that the population exhibited below-average SES compared to the overall population of Australia [1,2,9]. In previous studies conducted in Australia, populations with lower SES and lower health literacy had a poorer understanding of COVID-19 symptoms, did not view COVID-19 prevention protocols as important, and were more likely to endorse misinformed beliefs about COVID-19 and vaccinations [1,2,10].

The results of our study show that as SES increased, participants were more in agreement with the closure of the clinic during the first wave of COVID-19 (March 2020). They were more reluctant to attend dental appointments and book new appointments, stating that they would be less likely to attend dental appointments in the future if another outbreak occurred in the region. Furthermore, higher SES participants were more likely to report that the COVID-19-related restrictions put in place by the clinic were too lenient. Overall, as SES increased, the population was more fearful about COVID-19 and was less likely to attend the university clinic. These findings agree with that of McCaffery et al. in that higher SES populations who are more health-literate were more fearful and acknowledging of the virus compared to lower SES populations [10]. Higher SES has been noted to correlate with higher perceived values for oral health and attendance at dental health services [10]. However, patients may be more reluctant to attend dental services during COVID-19 because they believe contracting COVID-19 poses a greater risk to their overall health than missing dental appointments. Conversely, participants with a lower SES were less fearful, and COVID-19 did not seem to affect their attitudes or perceptions toward dental services. This may pose a problem in the case of a COVID-19 outbreak since chronic health conditions such as diabetes and hypertension, which are more commonly observed in lower SES populations, are associated with greater severity of illness and higher mortality rates from COVID-19 [11,12]. This would place lower SES populations in the regional areas at the greatest risk of fatalities if a COVID-19 outbreak were to occur.

Results from an analysis of access to dental services demonstrated that the lower SES population experienced more financial hardships due to COVID-19 than higher SES populations. This may be due to the specific industries of employment that were impacted in the region. According to data from the Australian Bureau of Statistics (ABS) in 2016, the main industry sectors in the region included accommodation, retail trade and cafés and restaurants [13], which were all significantly impacted by the lockdowns imposed [14]. Typically, higher SES populations were less affected in terms of health care and professional services [15]. Travel was also a factor that affected lower SES populations to a greater extent. These two factors may go hand in hand as those experiencing financial difficulties would likely have less readily accessible means for travel, especially from regional and remote areas.

Analysis of the results yielded no significant relationship between age and attitudes, perceptions, and access to dental services during the COVID-19 pandemic. A high proportion of participants were 70–79 years old (31.6%), and the majority attended the university clinic with pensioner concession cards. According to the Australian Government Department of Health, persons over the age of 70 are categorised as being at high risk of severe illness from COVID-19 [15]. This health advice has been widely accepted throughout Australia and globally. Studies conducted in countries such as the United States, which experienced more drastic outbreaks of COVID-19, show that as participants increased in age, their perceptions of the virulence of and vulnerability to contracting COVID-19 were higher [16]. They were less inclined to attend dental appointments compared to younger populations. In stark contrast, our study involving the region shows no significant trend to support the notion that older populations were more fearful of attending dental practices. This may be mostly attributed to the overall lower SES of the sampled population, who, as previously mentioned, are less fearful of COVID-19 due to the minimal impact the pandemic has had in the region in terms of the number of infected cases.

There are several limitations to this study. Although the recruited sample was random, the survey participants may not accurately represent adults residing in the region. Notably, the majority of participants were in the age group of 70–79 years, and most were pensioners [13]. According to the ABS, the median age in the region was 39 years in 2016, and only 15.2% of the population was above the age of 65, suggesting that the older age groups were over-represented in our sample [13]. Furthermore, our socio-economic indicators may not accurately represent the population, as census data (2016) show higher proportions of residents completing year 12 or further education compared to our study, in which most participants reported year 10 as being the highest education level attained [13]. Individuals with an Aboriginal or Torres Strait Islander background were represented at a percentage of 2.4% [13]. This is nearly half the expected representation in Queensland, which the ABS notes to be 4.6% [13,17].

This lower representation has been noted in previous research, and it is known that individuals with Aboriginal or Torres Strait Islander backgrounds are reluctant to attend health services [17,18]. It may also be argued that those who would have attended the university clinic when surveys were being conducted were those who were less fearful of COVID-19. More fearful people may not have attended the clinic and, thus, would not have had the opportunity to participate. This would have skewed the results to show that the region’s total population was less fearful than it may be. Furthermore, due to the fast-changing nature of the COVID-19 pandemic, new strains may pose renewed threats to the population. In the time between the planning of the study and the offer of surveys to participants, the delta strain became a more widespread and feared variant, and much speculation was happening regarding COVID-19 vaccinations. Currently, the milder but highly infectious omicron strain may significantly alter attitudes and perceptions towards oral health.

## 5. Conclusions

This study found that there is a statistically significant difference between socio-economic groups and their attitudes and perceptions of oral health care services in the current COVID-19 pandemic. Individuals from lower SES groups were less influenced by the pandemic. Participants from higher SES groups were found to be more cautious around COVID-19 and its risks. These differences could be remedied by increased public health education. Going forward, improvements could be made to the notification system at the university clinic so that patients receive information about clinic closures via text message services. Mandatory vaccination for university clinic employees and temperature checks were also highlighted to be important in assuring safety during the pandemic. Improving dental health services at locations such as university clinics is crucial as these clinics serve many underserved and neglected populations, such as the elderly and lower SES people. Understanding the impact of COVID-19 on these underserved populations is important to ensure that these members of our community are appropriately cared for and have their needs met.

## Figures and Tables

**Figure 1 healthcare-10-01747-f001:**
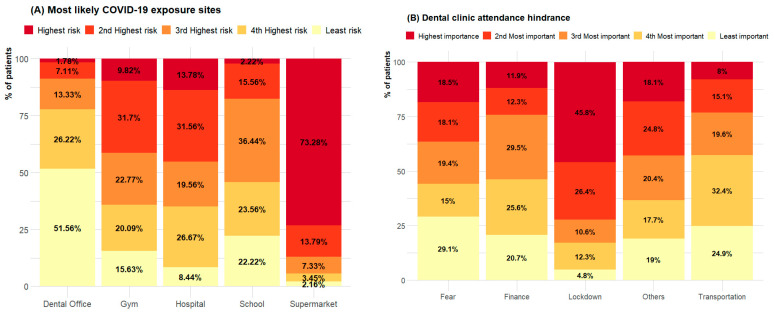
(**A**) Responses to question 9, patient’s perceived risk of exposure to COVID-19 at selected locations. (**B**) Responses to question 26, ranking of factors influencing patients attending a dental clinic.

**Figure 2 healthcare-10-01747-f002:**
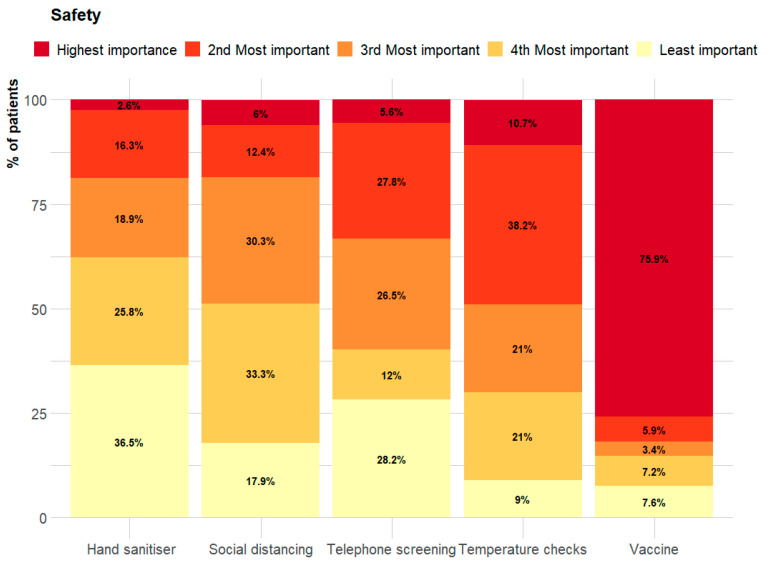
Reported actions that increase safety feelings of patients attending dental clinics during the COVID-19 pandemic.

**Figure 3 healthcare-10-01747-f003:**
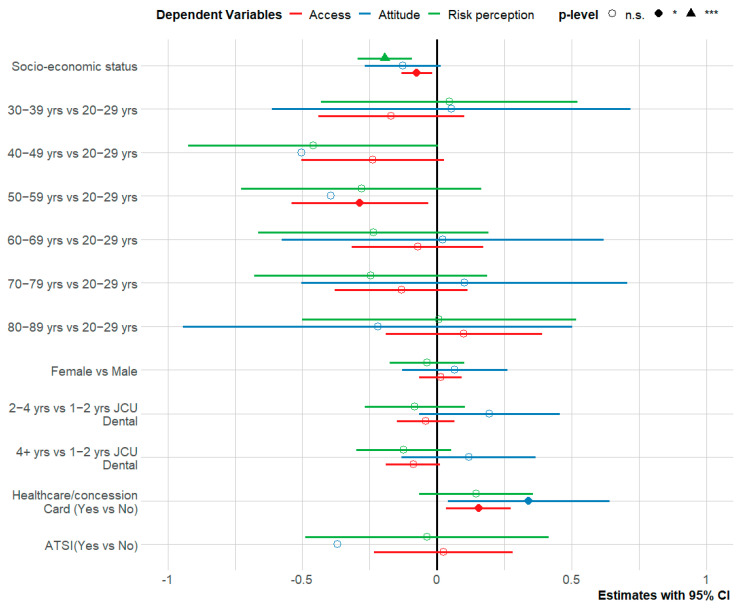
Effect of socio-demographic variables on risk perceptions, attitudes and access in multiple regression.

**Table 1 healthcare-10-01747-t001:** Socio-demographic characteristics of the study participants.

Variables	*n*	%
Overall	247	
Gender		
Male	122	49.4
Female	125	50.6
Age		
20–29	7	2.8
30–39	19	7.7
40–49	22	8.9
50–59	33	13.4
60–69	75	30.4
70–79	78	31.6
80–89	13	5.3
Years Attending JCU Dental		
1–2 Years	68	27.5
2–4 Years	74	30.0
4+ Years	105	42.5
Highest Level of Education		
Year 10	88	35.6
Year 12	54	21.9
Tafe/Certificate 3 or 4	45	18.2
Bachelor’s Degree	34	13.8
Post Graduate Degree	16	6.5
Annual Income Before Tax		
AUD <30,000	127	51.4
AUD 30,001–50,000	28	11.36
AUD 50,001–70,000	18	7.3
AUD 70,001–90,000	22	8.9
AUD 90,001–120,000	6	2.4
AUD >120,000	5	2.0
Employment Status		
Working Full Time	30	12.1
Working Part Time	41	16.6
Unemployed	29	11.7
Retired	136	55.1
Health Care or Concession Card		
Yes	184	74.5
No	55	22.3
Aboriginal or Torres Strait Islander		
Yes	6	2.4
No	234	94.7

**Table 2 healthcare-10-01747-t002:** Responses to perception questionnaire items regarding dental clinics during the COVID-19 pandemic.

	Significantly Better *n* (%)	Slightly Better *n* (%)	Much the Same *n* (%)	Slightly Worse *n* (%)	Significantly Worse *n* (%)	Mean	SD
How do you think the COVID-19 pandemic has impacted your dental health.	11 (4.5)	20 (8.1)	153 (61.9)	38 (15.4)	24 (9.7)	3.18	0.88
JCU Dental’s response to COVID-19 was appropriate; this included closure of the dental clinic in March 2020.	105 (42.5)	103 (41.7)	15 (6.1)	15 (6.1)	9 (3.6)	1.87	1.02
COVID-19 protocols implemented by JCU Dental included social distancing, mandatory patient handwashing and patient mouthwash performed prior to treatment.	21 (8.5)	26 (10.5)	176 (71.3)	14 (5.7)	10 (4)	2.86	0.81
Closing the JCU Dental in the March 2020 COVID-19 outbreak was appropriate.	154 (62.3)	38 (15.4)	21 (8.5)	21 (8.5)	13 (5.3)	1.79	1.22
If you have attended a dental clinic in the past 12 months, have the clinic’s COVID-19 protocols impacted fears around safety?	70 (28.3)	37 (15)	121 (49)	10 (4)	9 (3.6)	2.4	1.05

**Table 3 healthcare-10-01747-t003:** Responses to questionnaire items regarding access to dental clinics during the COVID-19 pandemic.

Access Score	Yes *n* (%)	No *n* (%)
Are you attending JCU Dental because you couldn’t attend your regular dentist?	200 (51)	40 (16.6)
COVID-19 pandemic has impacted my ability to travel when attending JCU Dental.	93 (37.6)	154 (62.3)
Since JCU Dental has re-opened after the COVID-19 pandemic, has the process changed to book an appointment?	101 (42.1)	139 (57.9)
Did you have an appointment cancelled due to the COVID-19 closure?	101 (42.1)	139 (57.9)
Did you have to attend another dental clinic because JCU Dental was closed due to COVID-19?	43 (17.4)	195 (78.9)
Were you notified of JCU Dental Clinic’s closure due to COVID-19?	106 (44.1)	133 (55.8)
Were you notified when JCU Dental Clinic planned to re-open?	117 (47.4)	120 (48.6)
Has COVID-19 caused you any financial hardship?	88 (38.3)	142 (61.7)
Has financial hardship impacted your ability to afford dental treatment?	40 (28.8)	27 (9.4)
Does the large number of patients attending the JCU Dental Clinic influence your attendance?	87 (35.6)	157 (63.6)

**Table 4 healthcare-10-01747-t004:** Responses to questionnaire items on attitudes regarding dental visits during the COVID-19 pandemic (responses are based on a 5-point Likert scale).

Access Score	Significantly Likely	Slightly Likely *n* (%)	Neither *n* (%)	Slightly Less Likely *n* (%)	Significantly Less Likely *n* (%)	Mean	SD
Has COVID-19 impacted your likeliness to attend the dental clinic?	30 (12.1)	29 (11.7)	138 (55.9)	19 (7.7)	28 (11.3)	2.94	1.07
Would you have scheduled an appointment if you had a dental emergency after the COVID-19 restrictions were put in place?	116 (47)	30 (12.1)	49 (19.8)	20 (8.1)	29 (11.7)	2.25	1.42
Are you likely to attend JCU Dental if we had a new outbreak of Cairns?	76 (30.8)	37 (15)	56 (22.7)	43 (17.4)	31 (12.6)	2.65	1.41
Does the large number of patients attending the JCU Dental Clinic influence your attendance?	17 (6.9)	20 (8.1)	157 (63.6)	23 (9.3)	27 (10.9)	3.09	0.94

**Table 5 healthcare-10-01747-t005:** Correlations analysis and internal consistency and reliability.

Indicators	1	2	3	4
1. Perceptions				
2. Access	0.473 ***			
3. Attitudes	−0.254 ***	−0.331 ***		
4. Socio-economic	−0.367 ***	−0.391 ***	0.354 ***	
Mean ± SD	2.42 ± 0.72	1.82 ±0.35	2.74 ± 0.93	1.90 ± 0.91
Cronbach’s α	0.649	0.67	0.61	0.665

*** *p* < 0.001.

## Data Availability

The data presented in this study are available upon request from the corresponding author. The data are not publicly available due to ethical reasons.

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
