# Peer review of "Impact of COVID-19 on Patients’ Attitudes and Perceptions of Dental Health Services: A Questionnaire Based Study in an Australian University Dental Clinic"

_healthcare, 2022, doi:10.3390/healthcare10091747_

Round 1

Reviewer 1 Report

In the Materials and Methods section there are several remarks:

- there is not any sign regarding the ethical considerations of the study

- it is not clear how paper based questionnaire was designed, on basis of which previous researches; also it is not clear how participants responded to the access scale domain-was it Likert scale, or something else

- it is a little bit confusing whether the authors wanted to determine normative characteristics of their questionnaire or not. If they did, why they only determined questionnaire reliability and not validity. Also, when Cronbach's alpha is calculated, this value is related to the reliability determined by its internal consistency. So, the phrase "internal consistency and reliability" is not appropriate at all. What about the validity of the questionnaire?

- it is also little bit confusing why the authors have performed Mann-Whitney and one-way ANOVA at the same time, because the one analysis excludes the other one.

- there is not any mention regarding multiple regression analysis in this section, although this analysis is presented later.

In the Results section there are several remarks:

- in the table 3 it is not clear with which variable the Mean and SD values are related to if the answers to the questions were Yes/No

-it is not clear how correlations in the Table 5 were performed (which method, test), and why this analysis was undertaken; further, internal consistency reliability analysis usually offers more data than presented-what is with other?

- it is not clear why multiple regression analysis was performed, due to the fact that there is not any mention of this analysis in the methodology section; also, how the regression model was designed? Usually regression analysis offers much more data that it was presented in the study, and offers to show prediction of available variables to some examined phenomenon. There is not anything about this in this study.

Author Response

Thank you for your comments. Plase see attached for our point to point responses.

Reviewer 2 Report

The authors sought to understand the impact on patients' attitudes and perceptions towards dental services against socio-economic status indicator, during Covid-19 pandemic outbreak. They used a cross-sectional study, using a questionnaire for data collection.

Abstract: Adequately presents the background, methodology, and main results. Two objectives are presented in the abstract: The study aimed to evaluate patients’ perception of risk associated with den-15 tal visits and explores appropriate methods to reduce this fear AND This study aimed to understand the 16 impact of COVID-19 on patients’ attitudes and perceptions of receiving dental health services at a 17 university clinic. Note that at the end of the introduction another objective is presented that does not coincide with any of those in the abstract: this research aimed to explore the relation between socio-economic status and COVID perception. Please clarify the objective, which should be consistent throughout the paper, as well as with the presentation of the results. The use of acronyms in the abstract should be avoided, but if used they should be preceded by the full expression (e.g. SES).

Key words: some of the key words are not MESH, and therefore should be reviewed.

Introduction: adequatly frames the research, however, it is not clear on the relevance of this study and how its results can be used. The authors state that evidence on the role of demographic variables in patients' perceptions of dental services is limited, but they do not clarify why this is relevant.

Methods: The rationale for the construction of the instrument should be presented. How were each of the items in each of the domains obtained/constructed? What is the framework for the construction of the instrument? Explain scoring system and interpretation. Explain socio-economic indicator (important variable for the results).

Results: results are hard to read. Please provide a concise description of the results. Do not repeat in the text, information that is already in the tables. Report the most relevant results related to the study's aim. Present % and n for all descriptive statistics. In table 2, 3 and 4 how was the mean obtained?

Discussion: adequate. Limitations are clearly stated.

Conclusion: even if optional, authors choose to make a conclusion. Having so, authors should highlight what these results had to current evidence and usefullness for clinical practice.

Author Response

(The authors gave the same response as above.)

Reviewer 3 Report

The article intitled “Impact of COVID-19 on Patients’ Attitudes and Perceptions of Dental Health Services” present the results of a survey submitted to patients during one month in an Australian dental clinic. The authors have obtained 247 answers. Their analyses are well performed but the validity of their results is limited to their own clinic. Moreover, it is difficult to consider suitable for publication in a 3.16 impact factor journal this type of short and non-previously validated survey. There are also many changes needed in the manuscript.

Comments on the manuscript:

-          The title does not fit the content of the article. It should be modified to be more informative concerning the objective or the results.

-          The first sentence of the abstract is unclear.

-          There are two objectives cited in the abstract, which is not possible.

-          Line 42: “chances” should be changed by “risks”

-          L57-60: the probability to develop a cellulitis from a pulpitis exists but it is not possible to say that someone with a pulpitis will have a cellulitis. There are too many factors that influence this evolution. The sentence must be modified.

-          L67-69: these elements are part of the discussion, not the introduction.

-          L70: if previous studies had already been performed, please indicate their references.

-          L71: same comment

-          L83-84: the inclusion of patients depending on their care history is unclear

-          Table 2: this table shows how the study interest is limited to JCU clinic.

-          In all the results (for example L143-156): please indicate n and %, not only % in the text.

-          L192-193: the α-values are acceptable but this classification means they are at the lower limit of acceptability. This should be mentioned in the discussion.

-          L194: the authors say that the socio-economic status is negatively correlated to Attitude. It is unclear, please mention the Table of Figure to refer for better understanding.

-          L207: maybe a word is lacking? “That is WHY”?

-          L229-230: need a reference

-          In the whole text, especially in the discussion, the references #1 and 2 are widely cited. Is it possible to find others publications to enlarge the discussion?

Author Response

(The authors gave the same response as above.)

Round 2

Reviewer 2 Report

The authors have made significant improvements to the manuscript.

Minor text editing issues need revison, eg, remove extra space in line 63 (study  performed), line 66 (the  gathering). 

Author Response

Thank you for the comments. We have assessed and corrected all the minor
editing issues including the line 63 and 66, as suggested. 

Reviewer 3 Report

The authors have performed many modifications in their document that clearly improves the quality of the manuscript. However, the general comments I sent for the first step of reviewing have not changed. The number of included subjects (247) is not very high in a questionnaire-based study, and the validity of this study’s results is limited to a dental clinic in Australia. It is thus difficult to consider suitable for publication in a 3.16 impact factor journal this type of survey-based article.

Author Response
Thank you for the comments. We do acknowledge that the participation numbers could have
been better. However, due to the timing of the project, just around the peak of COVID-19,
this is perhaps the best that could be expected from a single-centre study held in a clinic that
treated approximately 30 patients per day. Furthermore, the patients were just returning to
deal with non-urgent oral health issues and the patient attendance was much lesser than prepandemic levels. Since the data collection was paper-based and not an online survey, we feel
the responses collected (n=247) is significant. Consequently, we do consider the data
presented in this study addresses the gaps in the literature.
In regard to validity of results, COVID-19 has significantly affected the attitude, confidence
and health awareness whilst navigating through the pandemic and post-pandemic era, to a
varying degree across the world. Whilst the university dental clinic is a unique setting, the
outcome and trends noted are in line with the trends noted across the world (Example:
Abdulkareem AA et al Oral health awareness, attitude towards dental treatment, fear of
infection and economic impact during COVID-19 pandemic in the Middle East Int J Dent
Hyg 2021 Aug;19(3):295-304. doi: 10.1111/idh.12502).
Hence, we disagree with the reviewer’s comments in regard to ‘limited’ validity of results
reported in our study.